# Understanding the Inner Workings of Language Models Through Representation Dissimilarity

**Davis Brown**[1]     **Charles Godfrey**[2,†]     **Nicholas Konz**[1,3]
**Jonathan Tu**[1]     **Henry Kvinge**[1,4]
[1]Pacific Northwest National Laboratory     [2]Thomson Reuters Labs
[3]Duke University     [4]University of Washington
`{first.last}@pnnl.gov`

## Abstract

As language models are applied to an increasing number of real-world applications, understanding their inner workings has become an important issue in model trust, interpretability, and transparency. In this work we show that representation dissimilarity measures, which are functions that measure the extent to which two model's internal representations differ, can be a valuable tool for gaining insight into the mechanics of language models. Among our insights are: (i) an apparent asymmetry in the internal representations of model using SoLU and GeLU activation functions, (ii) evidence that dissimilarity measures can identify and locate generalization properties of models that are invisible via in-distribution test set performance, and (iii) new evaluations of how language model features vary as width and depth are increased. Our results suggest that dissimilarity measures are a promising set of tools for shedding light on the inner workings of language models.

## 1 Introduction

The defining feature of deep neural networks is their capability of learning useful feature representations from data in an end-to-end fashion. Perhaps ironically, one of the most pressing scientific challenges in deep learning is *understanding* the features that these models learn. This challenge is not merely philosophical: learned features are known to impact model interpretability/explainability, generalization, and transferability to downstream tasks, and it can be the case that none of these effects are visible from the point of view of model performance on even carefully selected validation sets.

When studying hidden representations in deep learning, a fundamental question is whether the internal representations of a given pair of models are similar or not. Dissimilarity measures (Klabunde et al., 2023) are a class of functions that seek to

---

† Work done at Pacific Northwest National Laboratory.

address this by measuring the difference between (potentially high-dimensional) representations. In this paper, we focus on two such functions that, while popular in computer vision, have seen limited application to language models: model stitching (Lenc and Vedaldi, 2014; Bansal et al., 2021) and centered kernel alignment (CKA) (Kornblith et al., 2019). *Model stitching* extracts features from earlier layers of model $f$ and plugs them into the later layers of model $g$ (possibly mediated by a small, learnable, connecting layer), and evaluates downstream performance of the resulting "stitched" model. Stitching takes a task-centric view towards representations, operating under the assumption that if two models have similar representations then these representations should be reconcilable by a simple transformation to such an extent that the downstream task can still be solved. On the other hand, CKA compares the statistical structure of the representations obtained in two different models/layers from a fixed set of input datapoints, ignoring any relationship to performance on the task for which the models were trained.

In this paper we make the case that dissimilarity measures are a tool that has been underutilized in the study of language models. We support this claim through experiments that shed light on the inner workings of language models: **(i)** We show that stitching can be used to better understand the changes to representations that result from using different nonlinear activations in a model. In particular, we find evidence that feeding Gaussian error linear unit (GeLU) model representations into a softmax linear unit (SoLU) model incurs a smaller penalty in loss compared to feeding SoLU activations into a GeLU model, suggesting that models with GeLU activations may form representations that contain strictly more useful information for the training task than the representations of models using SoLU activations. **(ii)** We show that dissimilarity measures can localize differences in models

that are invisible via test set performance. Following the experimental set-up of (Juneja et al., 2022), we show that both stitching and CKA detect the difference between models which generalize to an out-of-distribution test set and models that do not generalize. **(iii)** Finally, we apply CKA to the Pythia networks (Biderman et al., 2023), a sequence of generative transformers of increasing width and depth, finding a high degree of similarity between early layer features even as scale varies from 70 million to 1 billion parameters, an emergent "block structure" previously observed in CKA analysis of image classifiers such as ResNets and Vision Transformers, and showing that CKA identifies a Pythia model (`pythia-2.8b-deduped`) exhibiting remarkably low levels of feature similarity when compared with the remaining models in the family (as well as inconsistent architectural characteristics). This last finding is perhaps surprising considering the consistent trend towards lower perplexity and higher performance on benchmarks with increasing model scale seen in the original Pythia evaluations (Biderman et al., 2023) — one might have expected to see an underlying consistent evolution of hidden features from the perspective of CKA.

## 2 Background

In this section we review the two model dissimilarity measures appearing in this paper.

**Model stitching (Bansal et al., 2021):** Informally, model stitching asks how well the representation extracted by the early layers of one model can be used by the later layers of another model to solve a specific task. Let $f$ be a neural network and for a layer $l$ of $f$ let $f_{\leq l}$ (respectively $f_{\geq l}$) be the composition of the first $l$ layers of $f$ (respectively the layers $m$ of $f$ with $m \geq l$). Given another network $g$ the model obtained by stitching layer $l$ of $f$ to layer $m$ of $g$ with stitching layer $\varphi$ is $g_{>m} \circ \varphi \circ f_{\leq l}$. The performance of this stitched network measures the similarity of representations of $f$ at layer $l$ and representations of $g$ at layer $m$.

**Centered kernel alignment (CKA) (Kornblith et al., 2019):** Let $D = \{x_1, \ldots, x_d\}$ be a set of model inputs. For models $f$ and $g$ with layers $l$ and $m$ respectively, let $A_{f,l,g,m}$ be the covariance matrix of $f_{\leq l}(D)$ and $g_{\leq m}(D)$. Then the CKA score for models $f$ and $g$ at layers $l$ and $m$ respec-

tively and evaluated at $D$ is

$$\frac{||A_{f,l,g,m}||_F^2}{||A_{f,l,f,l}||_F ||A_{g,m,g,m}||_F}, \qquad (1)$$

where $|| \cdot ||_F$ is the Frobenious norm. Higher CKA scores indicate more structural similarity between representations. In our experiments we use an unbiased estimator of eq. (1) to calculate CKA in batches (see appendix C for details).

## 3 Model stitching reveals an asymmetry between GeLU and interpretable-by-design SoLU

The design of more interpretable models is an area of active research. Interpretable-by-design models often achieve comparable performance on downstream tasks to their non-interpretable counterparts (Rudin, 2019). However, these models (almost by definition) have differences in their hidden layer representations that can impact downstream performance.

Many contemporary transformers use the Gaussian error linear unit activation function, approximately $\text{GeLU}(x) = x * \text{sigmoid}(1.7x)$. The softmax linear unit $\text{SoLU}(x) = x \cdot \text{softmax}(x)$ is an activation function introduced in (Elhage et al., 2022) in an attempt to reduce neuron polysemanticity: the softmax has the effect of shrinking small and amplifying large neuron outputs. One of the findings of (Elhage et al., 2022) was that SoLU transformers yield comparable performance to their GeLU counterparts on a range of downstream tasks. However, those tests involved zero-shot evaluation or fine-tuning of the full transformer, and as such they do not shed much light on intermediate hidden feature representations. We use stitching to do this.

Using the notation from Section 2, we set our stitching layer $\varphi$ to be a learnable linear layer (all other parameters of the stitched model are frozen) between the residual streams following a layer $l$. We stitch small 3-layer (9.4M parameter) and 4-layer (13M parameter) SoLU and GeLU models trained by (Nanda, 2022), using the Pile validation set (Gao et al., 2020) to optimize $\varphi$.

Figure 1 displays the resulting stitching losses calculated on the Pile validation set. For both the 3-layer and 4-layer models, and at every layer considered, we see that when $f$ is a SoLU model for the stitched model $g_{>m} \circ \varphi \circ f_{\leq l}$ — the "SoLU-into-GeLU" cases — larger penalties are incurred than when $f$ uses GeLUs, i.e. the "GeLU-into-SoLU"

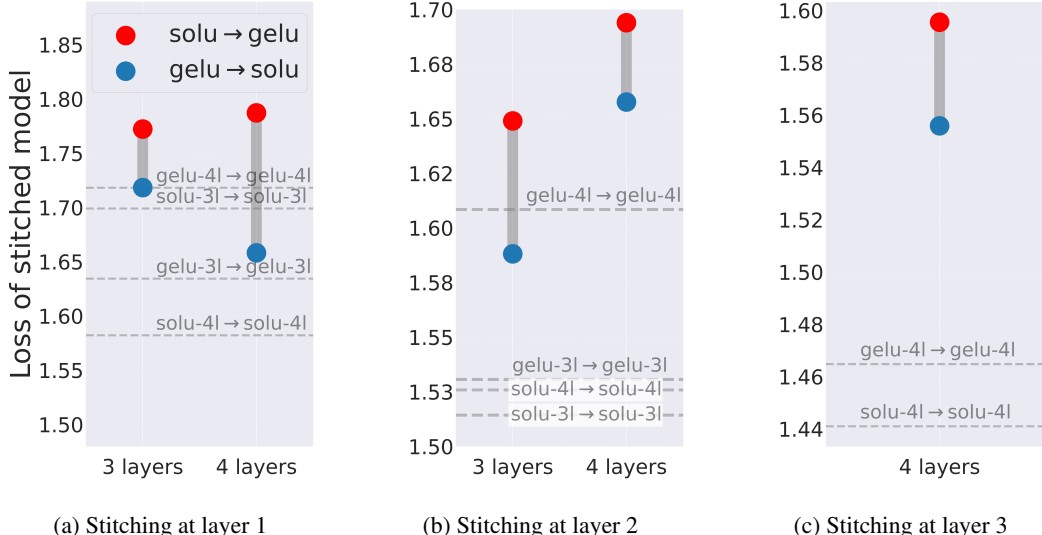

(a) Stitching at layer 1      (b) Stitching at layer 2      (c) Stitching at layer 3

Figure 1: **Comparing stitching loss** between 3-layer and 4-layer GeLU and SoLU models on the Pile validation set. Stitching for "solu→gelu," i.e. stitching with a SoLU 'head' and GeLU 'tail,' incurs systematic penalties (larger loss is worse) compared to the "gelu→solu" stitching. Baselines for stitching identical models (e.g., "gelu-3l→gelu-3l") are given to account for potential inherent differences in learning the stitching layer $\varphi$ between the activation functions.

cases. We conjecture that this outcome results from the fact that SoLU activations effectively reduce capacity of hidden feature layers; there may be an analogy between the experiments of fig. 1 and those of (Bansal et al., 2021, Fig. 3c) stitching vision models of different widths.

We also measure the stitching performance between pairs of identical models, the SoLU-into-SoLU and GeLU-into-GeLU stitching baselines. This is meant to delineate between two factors influencing the stitching penalties being recorded: the ease of optimizing stitching layers and the interplay between hidden features of possibly different architectures. We seek to measure differences between hidden features, while the former is an additional unavoidable factor inherent in model stitching experiments. The identical SoLU-into-SoLU and GeLU-into-GeLU stitching baselines serve as a proxy measure of stitching optimization success, since in-principle the stitching layer should be able to learn the identity matrix and incur a stitching penalty of 0. So, that SoLU-into-SoLU stitches better than GeLU-into-GeLU gives evidence for SoLU models being easier to stitch with than GeLU models, from an optimization perspective. That both SoLU-into-GeLU and GeLU-into-SoLU incur higher stitching penalties than GeLU-into-GeLU suggests that the penalties cannot only be a result of stitching layer optimization issues.

As pointed out in (Bansal et al., 2021) such an analysis is not possible using CKA, which only detects distance between distributions of hidden features (displayed for our GeLU and SoLU models in figs. 4 and 5), not their usefulness for a machine learning task. Further, the differences in layer expressiveness between GeLU and SoLU models are not easily elicited by evaluation on downstream tasks, but can be seen through linear stitching.

## 4 Locating generalization failures

Starting with a single pretrained BERT model and fine-tuning 100 models (differing only in random initialization of their classifier heads and stochasticity of batching in fine-tuning optimization) on the Multi-Genre Natural Language Inference (MNLI) dataset (Williams et al., 2018), the authors of (McCoy et al., 2020) observed a striking phenomenon: despite the fine-tuned BERTs' near identical performance on MNLI, their performance on Heuristic Analysis for NLI Systems (HANS), an out of distribution variant of MNLI, (McCoy et al., 2019) was highly variable. On a specific subset of HANS called "lexical overlap" (HANS-LO) one subset of fine-tuned BERTs (the *generalizing*) models) were relatively successful and used syntactic features; models in its complement (the *heuristic* models) failed catastrophically and used a strategy akin to

bag-of-words.[1] Recent work of (Juneja et al., 2022) found that partitioning the 100 fine-tuned BERTs on the basis of their HANS-LO performance is essentially equivalent to partitioning them on the basis of *mode connectivity*: that is, the generalizing and heuristic models lie in separate loss-landscape basins. We refer the interested reader to (Juneja et al., 2022) for further details.

But at what layer(s) do the hidden features of the generalizing and heuristic models diverge? For example, are these two subpopulations of models distinct because their earlier layers are different or because their later layers are different? Or both? Neither analysis of OOD performance nor mode connectivity analysis can provide an answer, but both stitching and CKA reveal that the difference between the features of the generalizing and heuristic models is concentrated in later layers of the BERT encoder. It is worth highlighting that conceptualizing and building relevant datasets for distribution shifts is often expensive and time consuming. That identity stitching and CKA *using in-distribution MNLI data alone* differentiate between heuristic and generalizing behavior on HANS-LO suggests that they are useful tools for model error analysis and debugging.

Figure 2a displays performance of pairs of BERT models stitched with the *identity function* $\varphi = \mathrm{id}$, i.e. models of the form $g_{>m} \circ f_{\leq l}$ on the MNLI finetuning task.[2] We see that at early layers of the models identity stitching incurs almost no penalty, but that stitching at later layers (when the fraction of layers occupied by at the bottom model exceeds 75 %) stitching *between* generalizing and heuristic models incurs significant accuracy drop (high stitching penalty), whereas stitching *within* the generalizing and heuristic groups incurs almost no penalty.

A similar picture emerges in fig. 2b, where we plot CKA values between features of fine-tuned BERTs on the MNLI dataset. At early layers, CKA is insensitive to the generalizing or heuristic nature of model A and B. At later layers (in the same range where we saw identity stitching penalties appear), the CKA measure *between* generalizing and heuristic models significantly exceeds its value *within* the generalizing and heuristic groups.

Together, the results of fig. 2 paint the following picture: the NLI models of (McCoy et al., 2020) seem to all build up generally useful features in their early layers, and only decide to memorize (i.e., use a lexical overlap heuristic) or generalize (use syntactic features) in their final layers.

## 5 Representation dissimilarity in scaling families

Enormous quantities of research and engineering energy have been devoted to design, training and evaluation of *scaling families* of language models. By a "scaling family" we mean a collection of neural network architectures with similar components, but with variable width and/or depth resulting in a sequence of models with increasing size (as measured by number of parameters). Pioneering work in computer vision used CKA to discover interesting properties of hidden features that vary with network width and depth (Nguyen et al., 2021), but to the best of our knowledge no similar studies have appeared in the natural language domain

We take a first step in this direction by computing CKA measures of features within and between the models of the Pythia family (Biderman et al., 2023), up to the second-largest model with 6.9 billion parameters. In all experiments we use the "deduped" models (trained on the deduplicated version of the Pile (Gao et al., 2020)). In the case of inter-model CKA measurements on `pythia-6.9b` we see in fig. 3 (bottom right) that similarity between layers $l$ and $m$ gradually decreases as $|m - l|$ decreases. We also see a pattern reminiscent of the "block structure" analyzed in (Nguyen et al., 2021), where CKA values are relatively high when comparing two layers *within* one of the following three groups: **(i)** early layers (the first 3), **(ii)** late layers (in this case only the final layer) and **(iii)** the remaining intermediate layers, while CKA values are relatively low when comparing two layers *between* these three groups.

Using CKA to compare features *between* `pythia-{1b,1.4b,2.8b}` and `pythia-6.9b` we observe high similarity between features at the be-

---

[1] By "relatively successful" we mean "Achieving up to 50% accuracy on an adversarially-designed test set." While variable OOD performance with respect to fine-tuning seed was also seen on other subsets of HANS, we follow (McCoy et al., 2020; Juneja et al., 2022) in focusing on HANS-LO where such variance was most pronounced.

[2] The motivation for stitching with a constant identity function comes from evidence that stitching-type algorithms perform symmetry correction (accounting for the fact that features of model A could differ from those of model B by an architecture symmetry e.g. permuting neurons, see e.g. (Ainsworth et al., 2023)), but that networks obtained from multiple fine-tuning runs from a fixed pretrained model seem to not differ by such symmetries (see for example (Wortsman et al., 2022)).

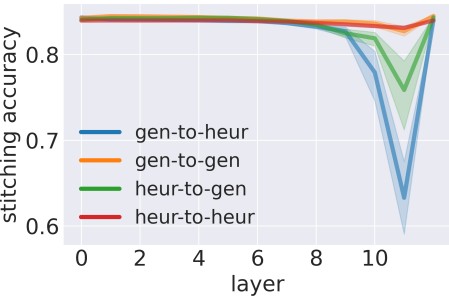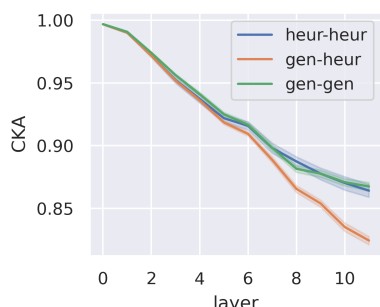

Figure 2: **Left:** Identity stitching on the MNLI dataset between pairs of the top 10 ("generalizing") and bottom 10 ("heuristic") performing models of (McCoy et al., 2020) on the lexical overlap subset of HANS (an out-of-distribution NLI dataset). **Right:** Corresponding CKA values. **Both:** Confidence intervals are obtained by evaluating on all distinct pairs of generalizing and heuristic models, for a total of $2 \cdot \binom{10}{2} + 10^2 = 190$ model comparisons.

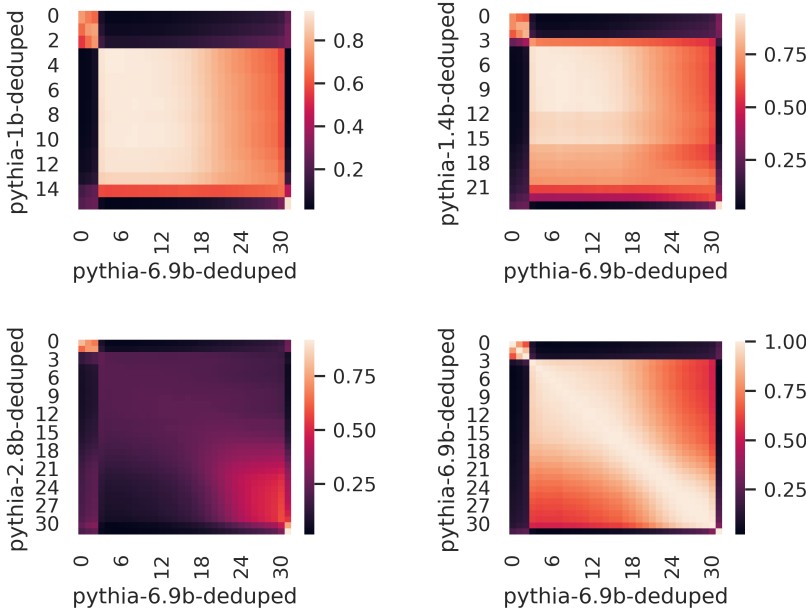

Figure 3: CKA between {pythia-1b, pythia-1.4b, pythia-2.8b, pythia-6.9b} and pythia-6.9b evaluated on the Pile dataset (Gao et al., 2020) (higher means more similar).

ginning of the model. A plausible reason for this early layer similarity is the common task of "detokenization," (Elhage et al., 2022) where early neurons in models may change unnatural tokenizations to more useful representations, for example responding strongly to compound words (e.g., birthday|party). We also continue to see "block structure" even when comparing between two models, and in the case of the pairs (pythia-1b, pythia-6.9b) and (pythia-1.4b, pythia-6.9b) substantial inter-model feature similarity in intermediate layers.

The low CKA values obtained when comparing intermediate layers of (pythia-2.8b, pythia-6.9b) break this trend – in fact, as illus-

trated in fig. 9, pythia-2.8b exhibits such trend-breaking intermediate layer feature dissimilarity with every Pythia model with 1 billion or more parameters.[3] Upon close inspection, we see that while the pythia-{1b, 1.4b, 6.9b} models all have a early layer "block" consisting of 3 layers with relatively high CKA similarity, in pythia-2.8b this block consists of only 2 layers, suggesting the features of pythia-2.8b diverge from the rest of the family very early in the model. In table 1 we point out that some aspects of pythia-2.8b's architecture are inconsistent with the general scaling trend of the Pythia family as a whole.

---

[3]Except itself, of course.

## Limitations

In the current work we examine relatively small language models. Larger models have qualitatively different features, and it is not obvious if our experiments on the differences in layer expressiveness between GeLU and SoLU models will scale to significantly larger models. While we show that identity stitching and CKA distinguish the heuristic and generalizing BERT models, we did not attempt to use stitching/CKA to automatically cluster the models (as was done with mode connectivity in (Juneja et al., 2022)).

## Ethics Statement

In this paper we present new applications of representation analysis to language model hidden features. Large language models have the potential to impact human society in ways that we are only beginning to glimpse. A deeper understanding of the features they learn could be exploited for both positive and negative effect. Positive use cases include methods for enhancing language models to be more safe, generalizable and robust (one such approach hinted at in the experiments of section 4), methods for explaining the decisions of language models and identifying model components responsible for unwanted behavior. Unfortunately, these tools can at times be repurposed to do harm, for example extracting information from model training data and inducing specific undesirable model predictions.

## Acknowledgements

This research was supported by the Mathematics for Artificial Reasoning in Science (MARS) initiative via the Laboratory Directed Research and Development (LDRD) investments at Pacific Northwest National Laboratory (PNNL). PNNL is a multi-program national laboratory operated for the U.S. Department of Energy (DOE) by Battelle Memorial Institute under Contract No. DE-AC05-76RL0-1830.

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

## A    Related Work

The problem of understanding how to compare the internal representations of deep learning models in a way that captures differences meaningful to the machine learning task has inspired a rich line of research (Klabunde et al., 2023). Popular approaches to quantifying dissimilarity range from shape based measures (Ding et al., 2021; Williams et al., 2021; Godfrey et al., 2022) to topological approaches (Barannikov et al., 2021).

The vast majority of research into these approaches has concentrated on computer vision applications rather than language models. Exceptions have looked at the evolution of model hidden layers with respect to a training task (Voita et al., 2019) or downstream tasks (Saphra and Lopez, 2019), statistical testing of sensitivity and specificity of dissimilarity metrics applied to BERT models (Ding et al., 2021), and more flexible representation dissimilarity measures aligning hidden features with invertible neural networks (Ren et al., 2023). While representation dissimilarity measures have not been extensively explored in language models, there is a rich line of work examining the structure of and relationship between hidden layers of NLP transformer models via probing for specific target tasks (Adi et al., 2016; Liu et al., 2019; Belinkov et al., 2017; Hewitt and Liang, 2019).

Our Section 4 borrows an experiment set-up from (Juneja et al., 2022), which found that partitioning BERT models using geometry of the fine-tuning loss surface (namely, linear mode connectivity (Frankle et al., 2019; Entezari et al., 2022)) can differentiate between models that learn *generalizing* features and *heuristic* features for a natural language inference task. Similar mode connectivity experiments for computer vision models were performed in (Lubana et al., 2022).

## B    Additional results

In figs. 4 and 5 we display CKA representation dissimilarity measurements for the GeLU and SoLU models used in the experiments of section 3. The purpose of these plots is simply to illustrate that in contrast to model stitching, CKA only detects distance between distributions of hidden features, not their usefulness for a machine learning task.

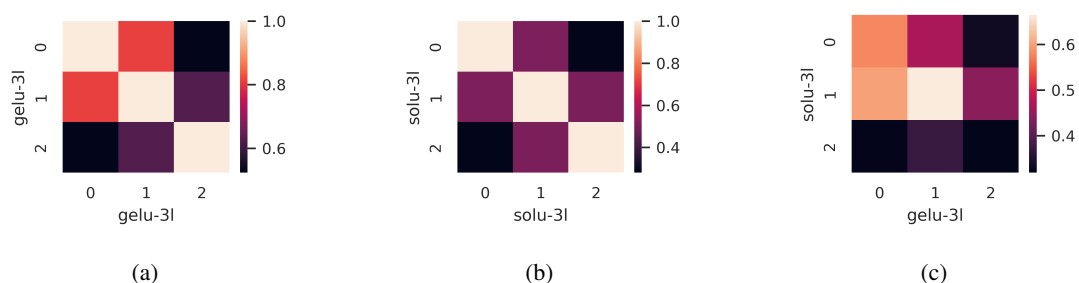

(a)                              (b)                              (c)

Figure 4: CKA for three layer GeLU and SoLU models (higher means more similar) evaluated on the Pile dataset (Gao et al., 2020). **(a, b)** Inter-model CKA for GeLU (respectively SoLU) models with 3 transformer blocks. **(c)** Between-model CKA for 3-block GeLU,SoLU models (rows correspond to SoLU layers, columns to GeLU layers

The plot in fig. 2 (left) shows identity stitching penalties for the MNLI fine-tuned BERT models on the MNLI dataset. In practice data distribution shifts are generally not known at the time of model development, and (as noted) the fact that identity stitching using MNLI data alone differentiates between heuristic and generalizing behavior on HANS-LO is potentially quite useful. Nevertheless, for comprehensiveness the performance of the identity stitched models on HANS-LO, displayed in fig. 7, also shows distinct differences between generalizing and heuristic models in the final stitching layer..

In fig. 2 (right) we display CKA between MNLI fine-tuned BERT models at each layer. This has the benefit of showing that the features of top (generalizing) and bottom (heuristic) models diverge in late BERT encoder layers (as we also saw with identity stitching in fig. 2 (left)). We aggregate the per-layer CKA measurements into a single *distance* measurement for each of the three cases (top-top, top-bottom, bottom-bottom) in fig. 8. This illustrates that the top (generalizing) and bottom (heuristic) models form

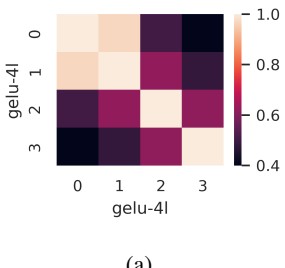 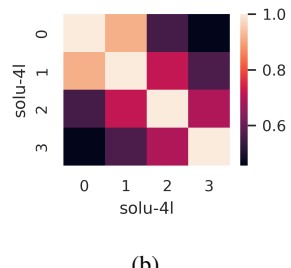 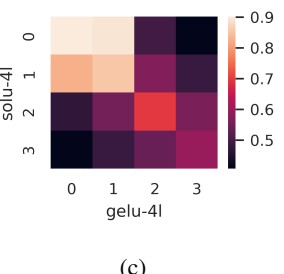

(a)            (b)            (c)

Figure 5: CKA for four layer GeLU and SoLU models (higher means more similar) evaluated on the Pile dataset (Gao et al., 2020). **(a, b)** Inter-model CKA for GeLU (respectively SoLU) models with 3 transformer blocks. **(c)** Between-model CKA for 3-block GeLU,SoLU models (rows correspond to SoLU layers, columns to GeLU layers.

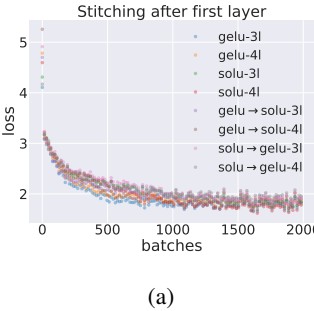 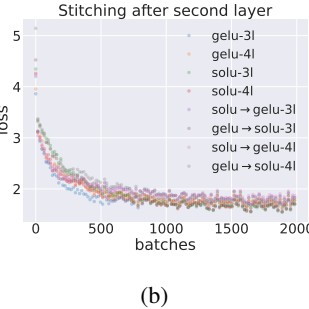 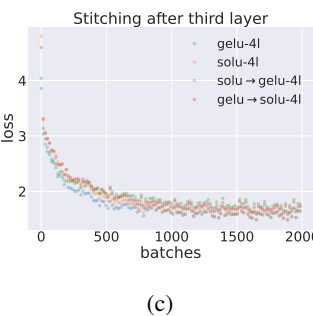

(a)            (b)            (c)

Figure 6: Learning linear stitching layers for the stitching GeLU and SoLU models in Figure 1.

two distinct clusters in a natural metric space of hidden features of MNLI datapoints; for the technical definition of this metric space we refer to appendix C.

In fig. 3 we saw high CKA values between intermediate features of `pythia-{1b,1.4b}` and `pythia-6.9b`, but comparatively *low* CKA values between intermediate features of the 2.8 and 6.9 billion parameter models. Figure 9 shows that more generally there is a high level of intermediate feature similarity for models in the set `pythia-{1b,1.4b,6.9b}` but a low level of intermediate feature similarity *between* `pythia-2.8b` and any of the models in `pythia-{1b,1.4b,6.9b}`.

Table 1 suggests one hypothesis for why `pythia-2.8b` possesses unique hidden features from the perspective of CKA, namely that its attention head dimension is quite a bit smaller than those of the `pythia-{1b,1.4b,6.9b}` models.

| model_name | d_head | rotary_dim |
|---|---|---|
| pythia-70m-deduped | 64 | 16 |
| pythia-160m-deduped | 64 | 16 |
| pythia-410m-deduped | 64 | 16 |
| pythia-1b-deduped | 256 | 64 |
| pythia-1.4b-deduped | 128 | 32 |
| pythia-2.8b-deduped | 80 | 20 |
| pythia-6.9b-deduped | 128 | 32 |
| pythia-12b-deduped | 128 | 32 |

Table 1: Select HuggingFace (Wolf et al., 2020) model configuration dictionary keys for the Pythia models, showing that the 2.8b model has an unexpectedly small attention head dimension and hence rotary embedding demension (the latter appears to be tied to 1/4 of the former).

We make a couple final notes on our analysis of Pythia models as it relates to prior work on image models. First, restricting attention to intra-model CKA heatmaps in fig. 10 we see an increase in intra-

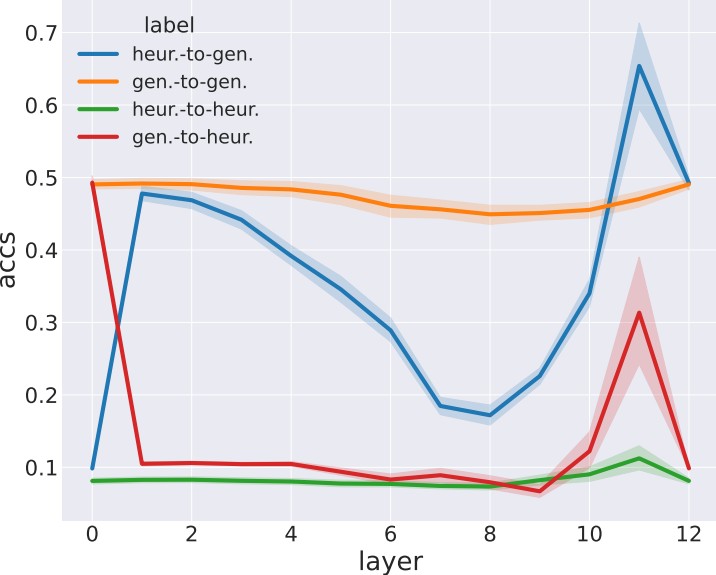

Figure 7: Identity stitching on the HANS-LO subset. The "heur.-to-heur." and "gen.-to-gen." models still exhibit distinct behavior.

model feature similarity as scale increases to 1 billion parameters,[4] analogous to an observation of (Nguyen et al., 2021). Note however that our experimental set up is not as clean as theirs: in our case both depth and width are scaled simultaneously, and perplexity of the Pythia models decreases as parameter count increases. In particular, our results would also be consistent with a hypothesis that intra-model feature similarity increases as validation loss decreases. Second, we speculate that the large blocks of intermediate features with high CKA similarity figs. 3 and 9 are related to the finding of (Raghu et al., 2022) that (compared to ResNets) Vision Transformers exhibit a high degree of feature uniformity across model layers. In other words, we hypothesize that this phenomenon is not specific to Vision Transformers, but rather transformer models more generally. However, as we only experiment with transformer models we are unable to make such a comparative statement and leave a broader study including e.g. RNNs and/or state-space sequence models to future work.

---

[4]See also relevant subplots of figs. 3 and 9 for larger model scales.

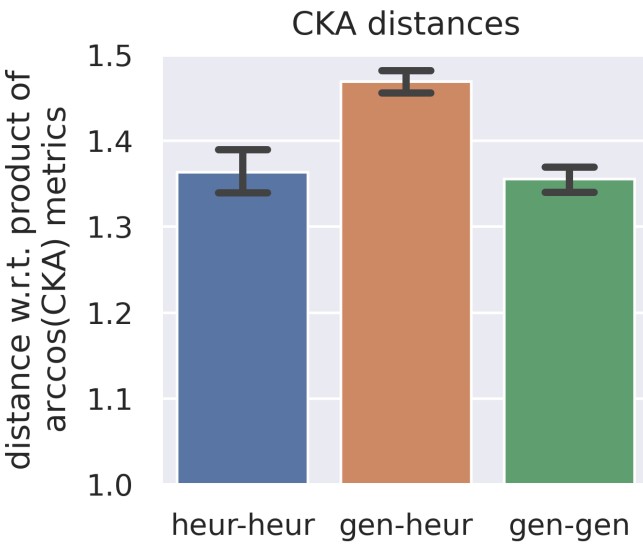

Figure 8: CKA-derived distances between the models of (McCoy et al., 2020) (higher means *less* similar, see appendix C for details on this distance metric). *Top-top*: pairwise CKAs between the top 10 models on HANS-LO. *Top-bottom*: pairwise CKAs between the top 10 and bottom 10 models on HANS-LO. *Bot-bot*: pairwise CKAs between the bottom 10 models on HANS-LO.

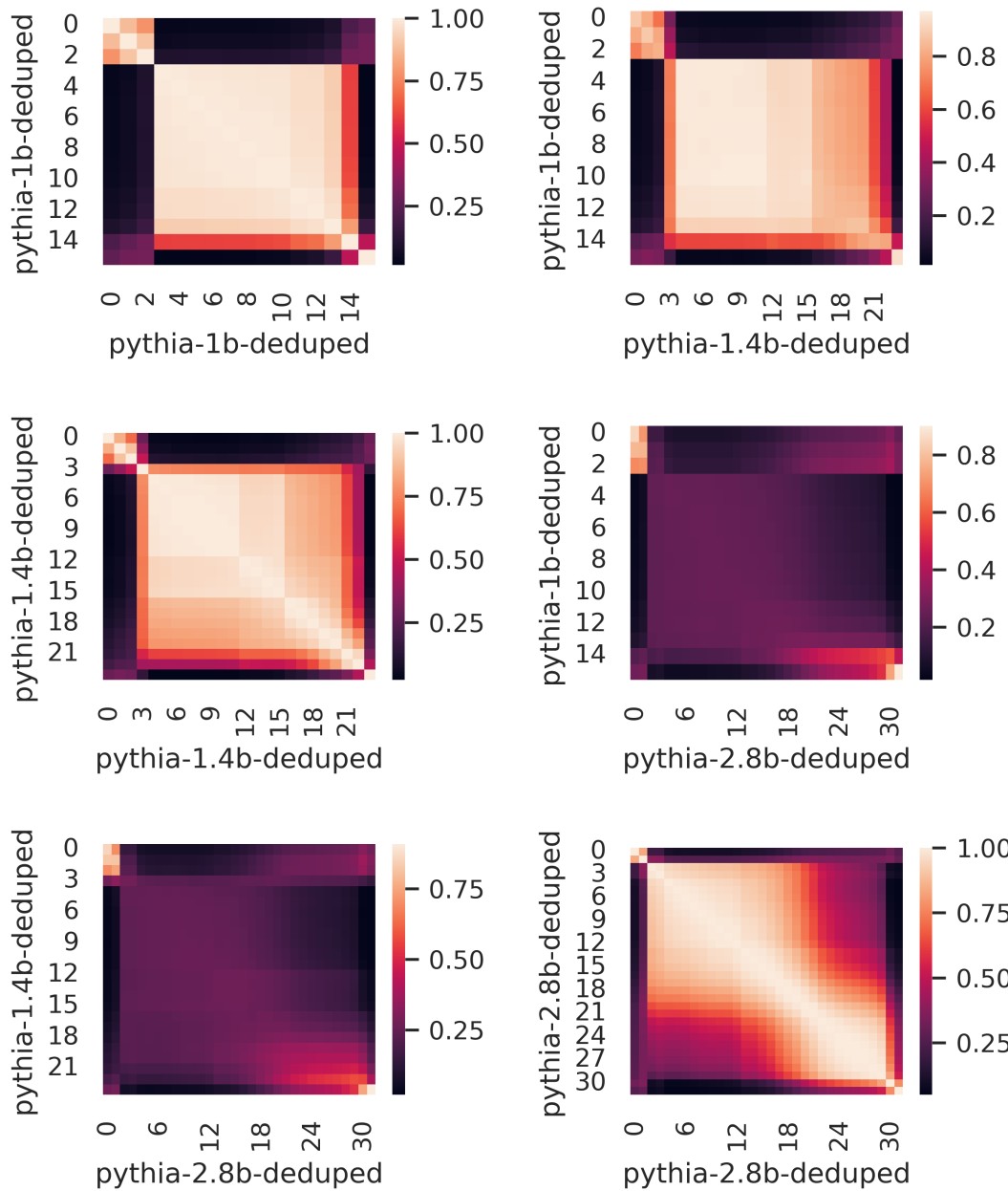

Figure 9: CKA for all pairs of models in the set `pythia-{1b,1.4b,2.8b}`, evaluated on the Pile dataset (Gao et al., 2020) (higher means more similar).

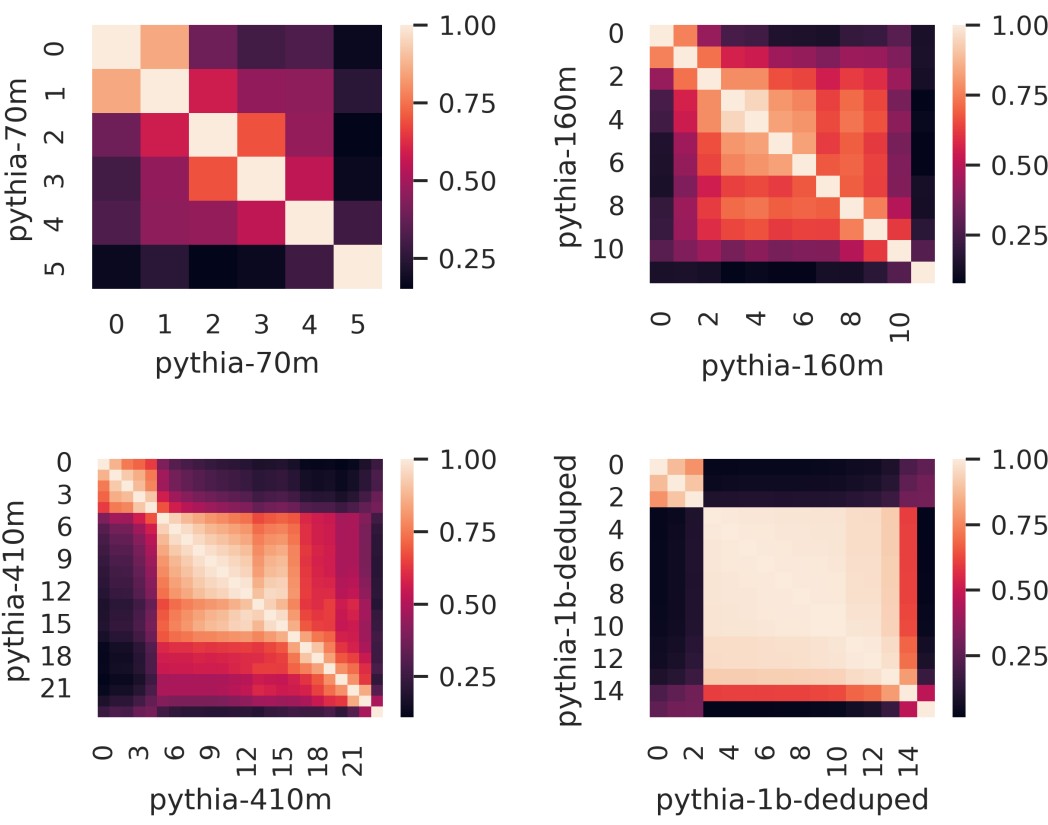

Figure 10: Intra-model CKA for models in the set `pythia-{70m,140m,410m,1b}`, evaluated on the Pile dataset (Gao et al., 2020) (higher means more similar).

## C   Experimental details

### C.1   SoLU/GeLU and Pythia models

For the SoLU-GeLU and Pythia experiments we use the TransformerLens library (Nanda, 2022), whose developers pre-trained the SoLU/GeLU models on the Pile (Gao et al., 2020) (architecture details can be found here and some training details are available here). We use a heavily adapted version of the tuned-lens library to performing stitching (Belrose et al., 2023). In particular, for stitching we learn an affine transformation (along with an additional LayerNorm) $\varphi$ with the PILE validation set. Figure 6 plots learning curves for GeLU-SoLU stitching layer optimization. We borrow many of the Tuned Lens hyperparameters, and train for 2000 steps (200 warm-up steps) with SGD with Nesterov momentum with a learning rate of 1.0 and cross-entropy loss.

When computing CKA, we extract features after each transformer block (*after* addition with the residual, technically the `hook_resid_post` hook point of TransformerLens). Letting $f$ and $g$ be the two models under consideration and letting $X$ be an input batch of token sequences, this yields feature tensors say

$$H_i = f_{\leq i}(X) \text{ and } H'_j = g_{\leq j}(X)$$

for $i = 1, \ldots, L$ and $j = 1, \ldots, L'$ where $L, L'$ are the respective depths of $f$ and $g$. In our experiments we use batches consisting of 1 (!) sequence of 1024 tokens (the default context length for the Pythia models); these sequences are constructed by "packing" datapoints from the Pile.[5] When one or more of $f, g$ is large (e.g. has more than 1 billion parameters), we run the necessary forward passes of $f$ and $g$ on separate GPUs — we used a mix of NVIDIA A100 GPUs with either 40 or 80 GB of RAM.

We then flatten the tensors of shape (batch, sequence, feature) (in our experiments (1, 1024, `hidden_dim`)) to a matrix of features with shape (batch * sequence, feature) (so in our experiments (1024, `hidden_dim`)),[6] and then compute CKA values in batches using an implementation of the unbiased estimator described in (Nguyen et al., 2021). Explicitly, if $H_i$ and $H'_j$ are (already flattened) hidden feature matrices of shape (batch * sequence, feature) (with possibly distinct feature dimensions), we first mean center both matrices ($H_i \leftarrow H_i - \mathbf{1}\mathbf{1}^T H_i$, similarly for $H'_j$) and compute the kernels

$$K_i = H_i H_i^T \text{ and } L_j = H'_j H'^T_j$$

Next, we make use of the unbiased Hilbert-Schmidt independence criterion $\text{HSIC}_1(A, B)$, defined for symmetric matrices $A$ and $B$ of equal shape $(n, n)$ as follows: first, let $\tilde{A}$ and $\tilde{B}$ be obtained by replacing the diagonals of $A$ and $B$ respectively with 0s.[7] Then,

$$\text{HSIC}_1(A, B) := \frac{1}{n(n-3)} \Big( \text{tr}(\tilde{A}\tilde{B}^T) + \frac{1}{(n-1)(n-2)} \text{tr}(\tilde{A}\mathbf{1}\mathbf{1}^T) \text{tr}(\tilde{B}\mathbf{1}\mathbf{1}^T) - \frac{2}{n-2} \tilde{A}\mathbf{1}\mathbf{1}^T \tilde{B} \Big).$$

This unweildy-looking expression can be computed efficiently by noting for example that $\text{tr}(\tilde{A}\tilde{B}^T)$ is simply the dot-product of the vectorizations of $\tilde{A}$ and $\tilde{B}$, $\text{tr}(\tilde{A}\mathbf{1}\mathbf{1}^T)$ is just the sum of all entries of $\tilde{A}$, and $\tilde{A}\mathbf{1}\mathbf{1}^T\tilde{B}$ is just the dot product of the vectors obtained by summing columns of $\tilde{A}$ and $\tilde{B}$.

Finally, we compute a matrix $S$ of shape $(L, L')$ with entries

$$S_{ij} = \frac{\text{HSIC}_1(K_i, L_j)}{\text{HSIC}_1(K_i, K_i) \cdot \text{HSIC}_1(L_j, L_j)},$$

and average these batch-estimated matrices $S$ over many randomly sampled batches, in our experiments 1000.

---

[5]Explicitly, we use the `chunk_and_tokenize` function from TransformerLens

[6]In other words, we are comparing activations at the token level. An alternative which we have not yet evaluated would be flattening the sequence and feature indices to a single dimension and comparing activations at the sample level.

[7]In PyTorch or NumPy, accomplished with $\tilde{A} = A - \text{diag}(\text{diag}(A))$.

## C.2 BERT models

For the fine-tuned BERTs experiments we use a fork of the codebase made available by (Juneja et al., 2022) (as well as their models made available on HuggingFace (Wolf et al., 2020)). When computing CKA values, we extract features after each transformer block (in this case before addition with the residual, as it was not immediately clear how to obtain features after the residual addition). The rest of our CKA calculation is very similar to that described for SoLU, GeLU and Pythia models above, with the following modifications: first, since MNLI is a classification task input sequences are padded to the maximum context length of the model, in this case 512, rather than "packed." Second, we collected features $H_i = f_{\leq i}(X)$ and $H'_j = g_{\leq j}(X)$ for many batches —in our case a total of $N \approx 1000$ datapoints — and stack them to obtain a matrix of shape ($N$, max length, feature) for each layer of each model. Then we flatten the first two indices to get matrices of shape ($N*$max length, feature), which are "chunked" along the first index — in our implementation to matrices of shape (1024, feature) — and passed to the batched, unbiased CKA estimator using $\text{HSIC}_1$ as above.

We do not expect these differences in implementation to make much difference, as what is described in the preceding paragraph is essentially equivalent to the procedure used for SoLU, GeLU and Pythia models bit with a batch size of 2 instead of 1.

To collapse layer-by-layer CKA to a single scalar value representing distance between the hidden features of two models with the same depth (as in fig. 8), letting $H_1, \ldots, H_L$ and $H'_1, \ldots, H'_L$ be the matrices of hidden features in the respective models and compute

$$\sqrt{\sum_{i=1}^{L} \arccos(\text{CKA}(H_i, H'_i))^2}. \tag{2}$$

The explanation of this expression is as follows: as pointed out in (Williams et al., 2021), while CKA is not a metric ($H = H'$ implies $\text{CKA}(H, H') = 1$, but for a metric it would have to be 0), the $\arccos$ if CKA is a metric in the mathematical sense. Equation (2) is then a metric on a product of hidden feature spaces obtained from the $\arccos$-CKA metric on each of the factors.