# OpenReview forum: "Understanding the Inner-workings of Language Models Through Representation Dissimilarity"
_EMNLP/2023/Conference — EMNLP 2023 Main_

### Official Review · Reviewer_ur3B · 2023-08-04

**Soundness:** 4

**Excitement:**

4: Strong: This paper deepens the understanding of some phenomenon or lowers the barriers to an existing research direction.

**Paper Topic And Main Contributions:**

The paper proposes to use dissimilarity metrics applied to hidden representation of different language models to derive insights into their inner workings.

**Questions For The Authors:**

1. For each of the study, could you elaborate on the potential impact of the phenomenon observed on the way models encode and process language?
2. Do you consider some of your observed phenomena to be unique to language models, or in general NLP models, or could they also apply for deep learning models in general?

**Reasons To Accept:**

The author(s) show how techniques used more commonly in e.g., computer vision, can be used to study NLP models.

For each of the three techniques presented, the methods and experiments are well described, with insights of different nature for each study.

**Reasons To Reject:**

The insights derived remain at an abstract machine learning level and the discussion remains rather disconnected from the fact that these are models specifically encoding and processing language. That is, the author(s) do not discuss the potential implications of these results on the way a LM encode and process a linguistic sequence, or the linguistic feature that it learns.

**Reproducibility:**

4: Could mostly reproduce the results, but there may be some variation because of sample variance or minor variations in their interpretation of the protocol or method.

**Reviewer Confidence:**

3: Pretty sure, but there's a chance I missed something. Although I have a good feel for this area in general, I did not carefully check the paper's details, e.g., the math, experimental design, or novelty.

---

> ### Author Rebuttal · Authors · 2023-08-29
>
> Thank you for your feedback and comments.
>
>  - “For each of the study, could you elaborate on the potential impact of the phenomenon observed on the way models encode and process language?”
>    - Our stitching/dissimilarity study for NLI models is perhaps worth highlighting. It was found that there are striking generalization differences in BERT models trained with different random seeds on NLI tasks, where some models learn a bag-of-words heuristic while others learn better generalizing syntactic heuristics [1]. Follow-up work demonstrated that such generalization differences can be identified without a dataset designed to exhibit domain shift (like the HANS-LO dataset) by examining linear mode connectivity [2]. We build on this result by demonstrating that the NLI models seem to build up generally useful features in their early layers, and only decide to memorize (i.e., use a lexical overlap heuristic) or generalize (use syntactic features) in their final layers. This is perhaps surprising as these modelling strategies appear to be quite difference and are in different loss basins [2]. We plan to add a description of the bag-of-words vs syntactic strategies when we cite [1, 2].
>    - We expect the differences between the SoLU and GeLU activation functions (if such generalization differences exist) to manifest as differences in how the transformer models encode and process language. Our experiments only demonstrate that such a difference may exist, and we leave further development to future work. However, we hope that our experiments indicate that model stitching is a useful preliminary tool for understanding how models encode language. Likewise, while our CKA experiments for the Pythia suite give evidence for the early layers of language models being similar across scales, we remain uncertain of the mechanistic behavior of the models. Nonetheless, we speculate that this is related to de-tokenization [3], where early layer MLPs may convert unnational token representations (e.g., separate tokens for a compound word “social” and “security") to representations more natural for downstream use (a single discrete representation for “social security”).
>
>  - “Do you consider some of your observed phenomena to be unique to language models, or in general NLP models, or could they also apply for deep learning models in general?” TL;DR our experiments are at least specific to sequence models, and we argue some are specific to NLP.
>    - The BERT NLI experiments test a generalization/robustness phenomenon that is specific to NLP.
>    - One technical aspect that we would further highlight in a camera-ready revision is that both stitching and CKA are conducted at the token level, i.e. we apply the same stitching matrix at every point in the sequence. Similarly, the features we apply CKA to are of size the hidden dimension of the transformer model in question, so we are comparing the similarity of representations at the token level, not just the sample level as is often done in other domains (like computer vision) – we outline the latter point in Appendix C. In this way, our experiments are at least specific to sequence models, and of course all models occurring in the paper are text models.
>    - Regarding our Pythia CKA evaluations, we hope that there is value in demonstrating that some phenomena observed in computer vision models can also be seen in LMs. Without running such experiments, the question of whether these are phenomena general to a class of neural networks regardless insensitive to the data they are trained on, or if they are domain specific, would remain open.
>
> [1] McCoy, R. T., Min, J., & Linzen, T. (2019). BERTs of a feather do not generalize together: Large variability in generalization across models with similar test set performance. arXiv preprint arXiv:1911.02969.
>
> [2] Juneja, J., Bansal, R., Cho, K., Sedoc, J., & Saphra, N. (2022). Linear connectivity reveals generalization strategies. arXiv preprint arXiv:2205.12411.
>
> [3] Elhage et al., “Softmax linear units,” Transformer Circuits Thread, 2022.

---

### Official Review · Reviewer_Dvu9 · 2023-08-04

**Typos Grammar Style And Presentation Improvements:** OOD is not defined in text (out of di…
**Soundness:** 4

**Excitement:**

4: Strong: This paper deepens the understanding of some phenomenon or lowers the barriers to an existing research direction.

**Paper Topic And Main Contributions:**

The paper introduces representation dissimilarity as a tool for investigating the inner workings of language models. The paper posits that understanding the inner workings of the models is necessary for interpreting their behavior. Three insights are provided in the paper to support this premise: 1) an apparent asymmetry in the internal representations of model using SoLU and GeLU activation functions; 2) evidence that dissimilarity measures can identify generalization properties of models that are invisible via in-distribution test set performance, and 3) new evaluations of how language model features vary as width and depth are increased. Two methods for representation dissimilarity are discussed: model stitching and centered kernel alignment.

**Reasons To Accept:**

The paper discusses a very relevant concern about language models: how different are language models and why do they (or not) generalize on out-of-distribution data.

**Reasons To Reject:**

Three insights are discussed in the paper. The insights on generalizability and scaling are understandable but the authors did not motivate the need for an in-depth study on activation functions.

**Reproducibility:**

3: Could reproduce the results with some difficulty. The settings of parameters are underspecified or subjectively determined; the training/evaluation data are not widely available.

**Reviewer Confidence:**

3: Pretty sure, but there's a chance I missed something. Although I have a good feel for this area in general, I did not carefully check the paper's details, e.g., the math, experimental design, or novelty.

---

> ### Author Rebuttal · Authors · 2023-08-29
>
> Thanks for your consideration of our submission and valuable comments.
>
> Regarding stitching models with different activation functions, our interest in SoLU layers stemmed from their interest from the point of view of neuron interpretability, superposition, and not-fully-understood effects of activation functions on hidden feature representations [1]. We realize that a set of experiments more directly inspired by related work in the computer vision literature (i.e. using models of varying parameter count/training time/training data size) would be worth carrying out as well (to our knowledge this would still be novel for LMs). In a longer paper we could of course expand this experiment suite considerably.
>
>
> [1] Elhage et al., “Softmax linear units,” Transformer Circuits Thread, 2022.

---

### Official Review · Reviewer_Mj3F · 2023-08-10

**Typos Grammar Style And Presentation Improvements:** Appendix B, Figure 4 & 5, "dblcheck -…
**Soundness:** 4

**Excitement:**

4: Strong: This paper deepens the understanding of some phenomenon or lowers the barriers to an existing research direction.

**Paper Topic And Main Contributions:**

This paper presents two ways to understand the language model's behavior by considering the models' dissimilarity. Specifically, they consider using model stitching to combine different but similar models, and CKA score analysis is also utilized to measure the layer-wise similarity between models.

**Questions For The Authors:**

A. If SoLU activation reduces the capacity of hidden feature layers compared to GeLU activation functions, why the solu-4l models perform better than gelu counterpart?

**Reasons To Accept:**

Two interesting methods are introduced and tried to analyze language models, despite using small models.

**Reasons To Reject:**

1. The clarity of the paper and the motivation of several design choices can be further improved. For example,
- Why only do model stitching on models with different activation functions?
- What's the Nanda, 2022 's model pre-trained on?
- (Line 158~160) For such a GeLU baseline, do you still put a linear stitching layer?
- What is HANS-"LO" different from HANS?
- Why the Figure 2's experiment only test on the lexical overlap subset of HANS?
- For experiments in Figure 2, are these figures do a ten times ten combination stitching /CKA score calculation for each line?

2. Overall, although I like the proposed methods, the conclusions we can draw from the paper is relatively sparse and not providing enough support. For example, if the author conjecture that SoLU activations effectively reduce the capacity of hidden feature layers, what is the corresponding influence? Is this asymmetry property correlate to final task performances? I suggest the author considering extend the paper to long paper and perform more thorough studies to support each conjecture.

**Reproducibility:**

3: Could reproduce the results with some difficulty. The settings of parameters are underspecified or subjectively determined; the training/evaluation data are not widely available.

**Reviewer Confidence:**

3: Pretty sure, but there's a chance I missed something. Although I have a good feel for this area in general, I did not carefully check the paper's details, e.g., the math, experimental design, or novelty.

---

> ### Author Rebuttal · Authors · 2023-08-29
>
> Thank you for your thorough read of our submission and suggestions for improvement.
>
> - “Why only do model stitching on models with different activation functions?” This is a valid question, and we think that a set of experiments more directly inspired by our references in the computer vision literature (i.e. using models of varying parameter count/training time/training data size) would be worth carrying out as well (to our knowledge this would still be novel for LMs). Our interest in SoLU layers stemmed from their interest from the point of view of neuron interpretability, their relevance to the phenomenon of superposition, and the not-fully-understood effects on hidden feature representations [1]. In a longer paper we would expand this experiment suite considerably.
> - “For such a GeLU baseline, do you still put a linear stitching layer?” Yes, we do.
> - “What is HANS-“LO” different from HANS? Why the Figure 2’s experiment only test on the lexical overlap subset of HANS?” HANS-LO is a particular subset of HANS testing whether an MNLI-trained model uses the (generally incorrect) heuristic “assume that a premise entails all hypotheses constructed from words in the premise” (see [2] and references therein). It would be interesting to expand our experiments to include the other 2 subsets, however we follow [3] in focusing on LO where the most striking generalization differences were observed in [2].
> - “For experiments in Figure 2, are these figures do a ten times ten combination stitching /CKA score calculation for each line?” Yes, exactly.
> - “… I suggest the author considering extend the paper to long paper and perform more thorough studies to support each conjecture.” We agree that a longer paper with more comprehensive experiments would provide deeper insights, and our decision to submit as a short paper was motivated by considerations very similar to what you outline here. Moreover, the conclusions of the BERT generalization experiments are more clear-cut than the SoLU-GeLU experiments you mention, identifying specific layers where feature-distortion is correlated with generalization performance. While we can continue to expand the paper and hope to build on these results in follow-up work, we felt that the results on hand were worth sharing.
>
> - R.e.: question on the stitching performance of SoLU-into-SoLU models, we include the stitching performance of identical SoLU-into-SoLU and GeLU-into-GeLU as a control for optimization differences for the linear stitching layer.  SoLU-into-SoLU stitches better than GeLU-into-GeLU, and this gives weak evidence for SoLU models being easier to stitch with than GeLU models, from an optimization perspective. That both SoLU-into-GeLU and GeLU-into-SoLU incur higher stitching penalties than GeLU-into-GeLU suggests that the penalties cannot only be a result of stitching layer optimization issues. We provide more context for our interpretation for this experiment in our response to reviewer rr9F above.
>
> [1] Elhage et al., “Softmax linear units,” Transformer Circuits Thread, 2022.
>
> [2] McCoy, R. T., Min, J., & Linzen, T. (2019). BERTs of a feather do not generalize together: Large variability in generalization across models with similar test set performance. arXiv preprint arXiv:1911.02969.
>
> [3] Juneja, J., Bansal, R., Cho, K., Sedoc, J., & Saphra, N. (2022). Linear connectivity reveals generalization strategies. arXiv preprint arXiv:2205.12411.

---

### Official Review · Reviewer_EMMQ · 2023-08-11

**Soundness:** 3

**Excitement:**

4: Strong: This paper deepens the understanding of some phenomenon or lowers the barriers to an existing research direction.

**Paper Topic And Main Contributions:**

This paper brings up a very important issue regarding the interpretability of deep neural networks. I can see their novelty is utilizing underrepresented techniques in NLP/NLU, e.g., model stitching and CKA, for such evaluation. Had I known more about this specific area, I would love to see these kinds of work published in EMNLP. However, I have to say I am not familiar with topics like this. I will defer the acceptance opinion to the other two reviewers. Wish the authors good luck!

**Questions For The Authors:**

1. I am not an expert in evaluating the activation functions so the following question is a clarification question: on p.1 in the last paragraph, there is an assumption that less penalty for loss leads to more useful information for the training task. Could you provide citations or footnotes explaining the rationale?

2. What is the theoretical contribution of your final step? p.2, apply CKA to the Pythia networks.

3. What is the theoretical rationale for choosing GeLU and SoLU as the particular model representations? You mentioned that most contemporary transformers are using them but are there more reasons?

After the rebuttal: All these questions are properly solved.

**Reasons To Accept:**

Novel techniques

**Reasons To Reject:**

None

**Reproducibility:**

3: Could reproduce the results with some difficulty. The settings of parameters are underspecified or subjectively determined; the training/evaluation data are not widely available.

**Reviewer Confidence:**

1: Not my area, or paper was hard for me to understand. My evaluation is just an educated guess.

**Typos Grammar Style And Presentation Improvements:**

### LaTeX style

1. Double check the in-text citation format. E.g., the experimental set up of (Juneja et al. 2022) -> the experimental set up of Juneja et al. (2022)

---

> ### Author Rebuttal · Authors · 2023-08-29
>
> Thanks very much for your thoughtful comments and questions.
>  - Re: “… there is an assumption that less penalty for loss leads to more useful information for the training task. Could you provide citations or footnotes explaining the rationale?” Our main reference here is [1] which found in a variety of experiments that stitching earlier layers of a model with more parameters/more training data/trained for more epochs (all things most practitioners generally feel lead to better internal representations) into later layers of a model with fewer parameters/fewer training data/trained for fewer epochs resulted in better performance than stitching in the opposite direction. We will add more discussion of this prior work as context.
>  - Re: contribution of CKA, using this tool to understand how architecture choices (CNNs vs. Vision Transformers) and model scale (in terms of width and depth) impact features of computer vision models. To our knowledge no similar studies have been conducted on language models. One can view our results as showing some phenomena previously observed in image classifiers can also be seen in LMs:
>       - An emergent block structure previously observed in CKA analysis of ResNets and ViTs [2], attributed to residual connections (see in particular Fig. 8). We will add a reference to [2] and this qualitative observation.
>       - Some evidence that as model width increases intra-model layer similarity increases (compare Figs. 3a and 8), parallel to another finding from [2]. Since submitting, we reworked our CKA evaluation to run on larger models (up to the 6.9 billion parameter Pythia model) and have some evidence that scaling width also increases inter-model layer similarity.
>       - Finally, the significant amount of similarity between the layers of a given Pythia model (seen in Fig. 8) is reminiscent of a finding on feature uniformity in ViTs in [3], although in our case we do not compare to other architectures. Using CKA to compare the representations of e.g., transformer models/self-attention layers to RNNs or state space models may be an interesting avenue for future work.
>  - Re: rationale for choosing the GeLU vs SoLU comparison. The SoLU activation function was designed to improve the neuron-level interpretability of transformer language models by reducing neuron superposition in MLP layers [1]. An interesting finding from [1] was that language models with the SoLU activation function achieved comparable performance to their GeLU counterparts in certain downstream evaluations. We hypothesized that the hidden feature representations (and maybe even generalization abilities) of small SoLU models are different (plausibly weaker) compared to their GeLU counterparts. Our stitching experiments support this hypothesis. However, we agree that this is only preliminary evidence, in contrast to the more straightforward conclusion from the BERT NLI study.
>
> [1] Bansal, Y., Nakkiran, P., & Barak, B. (2021). Revisiting model stitching to compare neural representations. Advances in neural information processing systems, 34, 225-236.
>
> [2] Nguyen, T., Raghu, M., & Kornblith, S. (2020). Do wide and deep networks learn the same things? uncovering how neural network representations vary with width and depth. arXiv preprint arXiv:2010.15327.
>
> [3] Raghu, M., Unterthiner, T., Kornblith, S., Zhang, C., & Dosovitskiy, A. (2021). Do vision transformers see like convolutional neural networks?. Advances in Neural Information Processing Systems, 34, 12116-12128.
>
> [4] Elhage et al., “Softmax linear units,” Transformer Circuits Thread, 2022.

---

### Official Review · Reviewer_rr9F · 2023-08-16

**Soundness:** 4

**Excitement:**

4: Strong: This paper deepens the understanding of some phenomenon or lowers the barriers to an existing research direction.

**Missing References:**

N/a

**Paper Topic And Main Contributions:**

This paper is an exploration of applying representation dissimilarity measures to a small selection of LMs and natural language tasks.

Specifically they use them to investigate the role that activation functions play, and the differences of representations captured in different layers of Pythia models of varying sizes. The dissimilarity functions they use are model stitching and centered kernel alignment (CKA).

The main 3 claims are:
1. model stiching works better for GeLU into SoLU vs SoLU into GeLU for LM
2. dissimilarity measures show differences not surfaced via test set performance
3. wider and deeper Pythia LMs have highly similar early layers across sizes, while intermediate layers grow more distant in representations

(1) matters because SoLU is designed to be interpretable, and assessing how this impacts learned reps as well as downstream performance are important. SoLU is intended to reduce polysemanticity using softmax rather than sigmoid as the nonlinear component in the activation function. While the authors who introduced it showed comparable SoLU performance to GeLU, they didn't analyze the representations.

The takeaway is that solu stitches together to itself better than gelu does, and *also* solu stitches on top of gelu better than gelu does on top of solu, when levels are held equal. The authors think this is because SoLU activations make lower-capacity layers. ~*I'm not sure if I buy this interpretation considering solu-solu stitches better than gelu-gelu?*~ **After the authors' rebuttal I am more convinced of this claim. The key point they make is that solu-solu vs gelu-gelu stitching baseline is intended to separate the inherent "stitchability" of the model from the utility of the features learned in the model for downstream adaptation. With this explanatory text added I think the camera ready will be more convincing to readers.**

(2) Is very important for a variety of questions, such as understanding the difference between generalizating and heuristic models for NLI by identifying divergent layers.

They show that MNLI test accuracy under model stitching alone can actually surface the difference between heuristic and generalizing models without deferring to HANS accuracy, just based on the marked dip in test accuracy performance when stitching in the high layers. This suggests that representations in the lower layers of a BERT model trained on MNLI contain consistent features, while the "decision" to memorize heuristics or learn the generalization features happens in the final few layers. Similarly, separation in CKA distance wrt layer between the classes of models also starts to occur at higher layers, further confirming this hypothesis.

(3) Is a bit more of an afterthought as it is introduced on the last page, but is a nice demonstration of the utility of CKA for analyzing LLM feature representations. In future work this could be expanded, and their narrow claim is demonstrated sufficiently by Figure 3.

**Questions For The Authors:**

- Rework figure 1 and clean up abbreviations and rethink terminology.
- Make the text in your subfigs larger (I couldn't read  Fig 1 until I zoomed in to 180%)

**Reasons To Accept:**

Timely demonstration of the utility of the 2 model analysis tools on a diverse set of problems

Overall this is a great example of an exploratory short paper that makes substantiated claims and will inspire future work.

Despite covering 3 very different applications of the two representation dissimilarities measures in only 4 pages, the authors do a good job of justifying their appropriateness and motivating them for unfamiliar readers. Overall I am impressed by how cohesive the story is in spite of its breadth and brevity

**I would like to see this paper accepted.**

**Reasons To Reject:**

~Some of the discussion of claim (1) feels a little shaky to me (see my summary).~ **The authors have addressed this concern in their rebuttal and I now agree. Hoping to see the camera ready changed in light of these suggestions.**

The rapid discussion of 3 wildly different application areas may feel hasty or too thin for some. This isn't a problem in my opinion.

**Overall I would not like to see this paper rejected**

**Reproducibility:**

5: Could easily reproduce the results.

**Reviewer Confidence:**

5: Positive that my evaluation is correct. I read the paper very carefully and I am very familiar with related work.

**Typos Grammar Style And Presentation Improvements:**

*I would not like to see this rejected, but I think there is some confusing writing in the early sections that needs to be addressed:*

related to claim (1):
- Lots of abbreviations like "resp." (eg, line 103) Is this wrt? respective? Idk.
- **Figure 1 really does not stand alone, and requires you to track through a bit of confusing double-negative logic to get the takeaways.** "larger is worse" phrasing for a value that can go negative (the performance penalty) is a little confusing because that can mean magnitude, I would frame it as "more positive is worse" or something. Additionally, the phrase "solu-gelu stitching (stitching with a SoLU 'head' and GeLu 'tail')" is very counterintuitive to me, as in LM we usually refer to upper layers as being the "head" of the model, but in this case the first activation function is the $f$ function which is on the bottom. It took me multiple rereads to parse this correctly, please reconsider phrasing.
Also, please explain what the red bars are in the caption for Figure 1 because you are overriding the de-facto standard "red is bad" connotation, which having a red bar in Figure 1 (a) be the most negative just made things really confusing.

---

> ### Author Rebuttal · Authors · 2023-08-29
>
> Thank you very much for your in-depth analysis of our results and valuable feedback.
>
> **Regarding interpretation of stitching results**: first, here is an outline of our thought process in slightly more detail. There are at least two factors influencing the stitching penalties being recorded – ease of optimizing stitching layers and the interplay between hidden features of possibly different architectures (the latter is what we seek to measure, the former is an additional unavoidable factor inherent in model stitching experiments). The identical SoLU-into-SoLU and GeLU-into-GeLU stitching baselines serve as a measure of stitching optimization success, since in principle the stitching layer should be able to learn the identity matrix and incur a stitching penalty of 0. So, that SoLU-into-SoLU stitches better than GeLU-into-GeLU gives weak evidence for SoLU models being easier to stitch with than GeLU models, from an optimization perspective. That both SoLU-into-GeLU and GeLU-into-SoLU incur higher stitching penalties than GeLU-into-GeLU suggests that the penalties cannot only be a result of stitching layer optimization issues.
>
> In this context, one way of unfolding your question is: it is unclear to us whether higher-capacity layers result in easier or harder stitching optimization. This is an interesting issue for us to follow up on; one could imagine that simpler features are easier to align, or alternatively since we are learning a linear layer that richer features yield a better-conditioned stitching optimization problem. While we might speculate that the stitching loss penalty is dominated by the downstream usefulness of the features rather than the optimization difficulty, we have not seen this question addressed in previous stitching work [1, 2] (or the intra-model stitching of [3]). Such optimization considerations, as well as other potential contributions to differences in stitching performance between SoLU-into-GeLU and GeLU-into-SoLU models, may change or modify our capacity interpretation. We will add language reflecting this experimental uncertainty and hope that our results may inform future approaches for specifying capacity or generalization differences (e.g., by guiding evaluation suites designed to identify performance differences in SoLU vs GeLU models).
>
> **Regarding presentation**:
>
> - Apologies, “resp.” was meant to denote “respectively.” We will update it and other ambiguous abbreviations where appropriate.
> - Regarding heads and tails, we will revise to be explicit here in the text (e.g. “stitching early layers of a SoLU model to later layers of a GeLU model”) and are considering using notation of the form “SoLU -> GeLU” in the plots to indicate the direction in which features are passed.
> - Regarding Figure 1, on rereading we empathize with your comments. To make the experiment more readable and remove unnecessary complications, we have 1) changed the stitching metric in the figure to be the absolute loss over the validation set (rather than the relative loss wrt the GeLU-into-GeLU stitched model), 2) will adopt “more positive is worse” phrasing, and 3) will rework the color scheme (and increase font size). To make the comparisons more obvious, we are also considering changing the bar plots to dumbbell plots to show the loss delta between SoLU -> GeLU vs GeLU -> SoLU (but maintaining the plot-per-stitching-layer scheme).
>
> [1] Bansal, Y., Nakkiran, P., & Barak, B. (2021). Revisiting model stitching to compare neural representations. Advances in neural information processing systems, 34, 225-236.
>
> [2] Csiszárik, A., Kőrösi-Szabó, P., Matszangosz, A., Papp, G., & Varga, D. (2021). Similarity and matching of neural network representations. Advances in neural information processing systems, 34, 225-236.
>
> [3] Belrose, N., Furman, Z., Smith, L., Halawi, D., Ostrovsky, I., McKinney, L., ... & Steinhardt, J. (2023). Eliciting latent predictions from transformers with the tuned lens. arXiv preprint arXiv:2303.08112.

---

### Meta-Review · Area_Chair_Nqcd · 2023-09-19

**Recommendation:** 5

**Metareview:**

This work explores using dissimilarity measures for model interpretability, including findings that these measures can give insights about model generalization. All reviewers appreciated the execution and motivation of this work, and it would be a good short paper at EMNLP.

---

### Decision · Program_Chairs · 2023-10-07

**Decision:**

Accept-Main

**Comment:**

This work explores using dissimilarity measures for model interpretability, including findings that these measures can give insights about model generalization. All reviewers appreciated the execution and motivation of this work, and it would be a good short paper at EMNLP.